# Evolution of Influenza Viruses—Drug Resistance, Treatment Options, and Prospects

**DOI:** 10.3390/ijms232012244

**Published:** 2022-10-13

**Authors:** Julia M. Smyk, Natalia Szydłowska, Weronika Szulc, Anna Majewska

**Affiliations:** Department of Medical Microbiology, Medical University of Warsaw, Chalubinskiego 5 Str., 02-004 Warsaw, Poland

**Keywords:** anti-influenza drugs, influenza treatment, drug resistance

## Abstract

Viral evolution refers to the genetic changes that a virus accumulates during its lifetime which can arise from adaptations in response to environmental changes or the immune response of the host. Influenza A virus is one of the most rapidly evolving microorganisms. Its genetic instability may lead to large changes in its biological properties, including changes in virulence, adaptation to new hosts, and even the emergence of infectious diseases with a previously unknown clinical course. Genetic variability makes it difficult to implement effective prophylactic programs, such as vaccinations, and may be responsible for resistance to antiviral drugs. The aim of the review was to describe the consequences of the variability of influenza viruses, mutations, and recombination, which allow viruses to overcome species barriers, causing epidemics and pandemics. Another consequence of influenza virus evolution is the risk of the resistance to antiviral drugs. Thus far, one class of drugs, M2 protein inhibitors, has been excluded from use because of mutations in strains isolated in many regions of the world from humans and animals. Therefore, the effectiveness of anti-influenza drugs should be continuously monitored in reference centers representing particular regions of the world as a part of epidemiological surveillance.

## 1. The Evolution of Influenza Viruses

Influenza viruses are members of the *Orthomyxoviridae* family. They are enveloped, segmented, single-stranded, and negative-sense RNA (-ssRNA) viruses. Based on the antigenic differences in the nucleoprotein (NP) and matrix protein (M), influenza viruses are divided into four genera: Alphainfluenzavirus (species: influenza A virus; IAV), Betainfluenzavirus (species: influenza B virus; IBV), Gammainfluenzavirus (species: influenza C virus; ICV), and Deltainfluenzavirus (species: influenza D virus; IDV) [1,2]. In 1933, the first human influenza virus was isolated: IAV [3]. In the following decade, two more influenza viruses, the IBV and the ICV, were discovered [4]. Finally, in 2011, the newest influenza genus was isolated, and later, in 2016, it was officially named the IDV [2,5].

IAV has the greatest clinical significance among influenza viruses, as it causes annual epidemics in the human population. This zoonotic pathogen can circulate in a variety of avian and mammalian species (both among wild and farmed animals). IAV’s ability to jump between its hosts is one of the reasons for its evolutionary success and it can occasionally result in pandemic outbreaks [6,7,8].

IAV is divided into subtypes based on the type of their surface glycoproteins: haemagglutinin (HA) and neuraminidase (NA). Currently, there are eighteen known HA subtypes (H1–18) and eleven neuraminidase NA subtypes (N1–11) [9]. The H17N10 subtype was discovered in Central America (Guatemala, 2009), whereas the H18N11 subtype was first encountered in Peru (2010). Both were isolated from rectal swabs of flat-faced fruit bats. Bats are an important reservoir host for numerous other viruses, such as coronaviruses, filoviruses, henipaviruses, and lyssaviruses [10]. Previous studies have reported that H17N10 and H18N11 subtypes, found in bats, cannot replicate in vitro in both human and mammalian cell lines. H17 and H18 HAs do not bind (or bind weakly) to sialic acids, which are known as canonical receptors of IAVs. Moreover, N10 and N11 NAs are genetically distant from molecules found on IAV (N1–N9) [11,12].

According to the latest research, avian and mammalian cell lines are susceptible to bat-derived influenza viruses, but their biological properties are still unknown. Both H17N10 and H18N11 subtypes can infect ferrets and mice, although their replication capability is at a low level [13]. There are only three combinations of HA and NA that circulate among humans: A/H1N1, A/H2N2, and A/H3N2 [9].

In comparison to IAV, the IBVs are far less genetically diverse, and their mutation rate is two to three times slower. Apart from humans, IBVs can be rarely isolated from animal reservoirs, such as seals, pigs, and ferrets. Since IBVs have such limited animal reservoirs, they have no pandemic potential. There are currently two lineages of IBV, B/Yamagata and B/Victoria, circulating the globe [4,14,15].

Influenza C viruses (ICVs) are isolated in all parts of the world. Their primary reservoir is humans. ICVs usually cause mild symptoms and affect mostly children from the age one to six. This virus can also occasionally infects pigs and dogs. As for IDVs, their primary reservoir is cattle; however, they can also spread to other mammalian species, including pigs, goats, and sheep. The current knowledge of IDVs is still limited, and there is no conclusive evidence of their ability to infect humans [2,4,16].

The influenza virus produces pleomorphic virions which can take a spherical, bacillus, or filamentous shape. However, laboratory-derived isolates typically adapt spherical virions with a mean outer diameter of 120 nm [17]. The virion consists of the genome enclosed by the viral matrix proteins, which are surrounded by an outer layer of a lipid bilayer membrane, called the viral envelope. This protective layer is formed from the plasma membrane of the host cell. The viral IAV and IBV genomes comprise eight segments, but ICV and IDV comprise seven segments, of a single-stranded, negative-sense RNA bound by the viral ribonucleoprotein complexes [2,5,18,19].

Thus far, out of all the influenza viruses, the genome of IAV has been studied the most since it poses the greatest threat to public health and is the primary target of antiviral drugs. The IAV genome encodes up to 17 proteins; the most important include:RNA polymerase subunits (PB1, PB2, and PA), which form the multifunctional viral RNA-dependent RNA polymerase (RdRP) and are involved in the processes of replication and transcription.PB1-F2 protein, which participates in cell apoptosis.PA-X protein, which is a virulence factor. It selectively degrades host mRNAs, leading to protein shutoff.Hemagglutinin (HA), which is a receptor-binding and membrane fusion glycoprotein.NP nucleoprotein, which is a structural protein encapsulating the negative strand of the viral RNA.NA (neuraminidase), which facilitates the release of new virus particles from the host cell.Two matrix (M1 and M2) proteins, which play key roles in the assembly and release of new viral particles. The enveloped M2 protein forms a proton selective ion channel, a target for anti-influenza drugs.Two distinct nonstructural proteins (NS1 and NS2) that are synthesized in infected cells and play an important role in the translation process. NS2 mediates the nuclear export of viral ribonucleoproteins [14,15].

The structure of the IAV and IBV is so similar that they are indistinguishable under the electron microscope. IAV has three types of surface proteins: HA and NA, in a ratio of around four to one, and a few transmembrane proteins forming ion channels in the viral M2, which acts as a proton-selective viroporine and plays a part in the uncoating and assembling of the virus particles. IBV’s envelope is also studded with HA and NA proteins [20]. In comparison, the ICV and IDV have only one major envelope glycoprotein, called the hemagglutinin-esterase-fusion (HEF) protein. The HEF protein possesses receptor binding, receptor destroying, and membrane fusion activities. Therefore, it is a functional equivalent of HA and NA of IAV and IBV. Apart from the HEF protein, ICV encodes only two other membrane proteins, CM1 (matrix protein) and CM2 (short tetrameric membrane glycoprotein), structurally analogous to the M2 protein in IAV [21,22,23].

RNA viruses have a high mutation rate because of the error-prone replication feature of the RNA-dependent RNA polymerase (RdRp). Incorporating the wrong nucleotides occurs with a frequency of 10^−3^–10^−4^ per replication cycle. The lack of autocorrective properties of the RdRp leads to variability in influenza viruses. They gain an antigenic diversity via two main mechanisms: antigenic drift and shift. Antigenic drift is a continuous process which results from a minor changes in the RNA segments encoding the surface antigens, HA and NA. This mechanism enables the virus to escape from antibody-mediated neutralization acquired after infection or vaccination [6,24]. Antigenic shift is a result of the reassortment of gene segments between two or more distinct strains of influenza viruses following their coinfection of the same host (i.e., bird or swine). Such reassortment has evolutionary importance because it results in the emergence of new genomic variants. This mechanism was crucial for the appearance of highly pathogenic avian influenza viruses (HPAV), such as H5Nx and H7N9, which caused infections in humans, and the A/(H1N1)pdm09, responsible for the pandemic in 2009.

In theory, the gene reassortment might lead to production of as many as 256 genetically different types of viral progeny. A new variant is unrecognizable by the host’s defense mechanisms. If it replicates effectively, it may even cause a pandemic [6,25].

The 20th century has brought about three influenza pandemics. In 1918, the A/H1N1 strain began to circulate. Its appearance was a result of the reassortment of human and avian strains. The so-called Spanish flu (because Spain remained neutral during World War I, the press, free from war censorship, could report on many victims of the disease) began in the USA and lasted until 1920. It was estimated that during this time, 500 million people (mostly aged 25–40) experienced symptoms of the Spanish flu [26,27].

In early 1957, a new variant, A/H2N2, was identified in China (Asian influenza). Three segments of its genome, PB1, HA, and NA, were derived from a strain found in a wild duck. Its mortality rate was significantly lower (2.2%) compared to the Spanish flu (10%), thanks to the quick identification of the virus and the availability of ready-to-use flu vaccines [24].

In 1968, a new A/H3N2 influenza subtype emerged and caused yet another pandemic, the Hong Kong influenza. The A/H3N2 virus contained two genes derived from low-pathogenicity avian influenza (LPAI) A virus and six genes from the A/H2N2 that had been circulating among people since its emergence in 1957, when they resulted in an A/H2N2 pandemic. The milder symptoms and the lower mortality rate were related to the availability of the vaccine and the first anti-influenza drugs [28,29,30].

Some experts emphasize the importance of the 1977 influenza outbreak and consider it to be the fourth influenza pandemic in the XX century. Colloquially, it was referred to as the Russian flu since the former USSR was the first country to report its outbreak. The infection was caused by the A/H1N1 virus and affected mainly young people (26 years of age and younger). This resulted from the fact that the virus was not novel. A similar strain caused infections in the past (between 1947 and 1957). That gave a chance to the adult population to develop antibodies against this virus, resulting in a lower death rate (<5/100,000) [31].

In April 2009, a new variant, A/H1N1, appeared in Mexico and California and was responsible for the first pandemic of the XXI century. It was a quadruple reassortant consisting of two swine-origin viruses, one avian-origin virus, and one human-origin virus. This new virus, named A/(H1N1)pdm09, turned out to be significantly different from the classic seasonal influenza A/H1N1 viruses. The average age of the individuals affected was approximately 18 years (64% of the cases occurred among people aged 10 to 19). The fatality rate among the laboratory-confirmed cases was 2.9% [30].

## 2. Treatment Options and Emerging Limitations in the Treatment of Influenza

Johns Hopkins Center for Health Security in the USA has worked on analyzing the characteristics of microorganisms that could render them capable of becoming a global catastrophic biological risk (GCBR). The influenza virus meets all the criteria for such a pathogen, namely:It is transmitted through the respiratory route;It efficiently spreads from person to person;It can be found in the nasopharyngeal secretions of an infected person;It causes mild infections, and does not lead to death in every case;It causes a significant case fatality ratio;It can evade the host immune system (genetic variability);It causes an infection requiring specialized medical equipment for its treatment (availability of this equipment is limited) [32].

For those reasons, over the years, numerous studies have been conducted to develop effective anti-influenza medicines. It is necessary to create a broad spectrum of antiviral medicines with a high genetic barrier to resistance, which will be effective against multiple strains of the virus to counteract the risk of selection of resistant strains.

Currently available antiviral drugs reduce the duration of influenza symptoms, thus limiting the transmission of the virus and reducing the influenza mortality risk. However, because of the possibility of selecting resistant strains, influenza treatment remains a challenge, and its effectiveness should be monitored [33].

### 2.1. M2 Protein Inhibitors (M2Is)

The M2 protein inhibitors include amantadine (Symmetrel^®^), synthesized in the 1960s, and rimantadine (Flumadine^®^), discovered two decades later. Initially, both drugs were effective in treating and preventing infection caused by IAV, with efficacy rates reaching up to 90%. M2Is could not be used to treat influenza B infection because IBV does not have the M2 protein on its surface.

M2Is act by blocking the hydrogen ions influx through the M2 ion channel into the viral interior, therefore stopping the M2-related pH lowering. As a result, they prevent the viral uncoating and the subsequent release of viral RNA into the host cell. M2 protein inhibitors are derived from adamantane, which is a symmetric tricyclodecane made of three fused cyclohexane rings. They adopt a chair conformation and form a cage-like structure. The success of the treatment depends on an early administration of the drugs, preferably within 48 hours from the influenza symptoms onset. M2 inhibitors were also recommended as a short-term prophylaxis for individuals who were living with infected people, especially if they had an increased risk of influenza-related complications.

The resistance of the virus to amantadine was first observed during the 1980 epidemic. It remained at the low level of 1–2% until the year 2000. From 2000 to 2004, amantadine resistance among the A/H3N2 subtype increased from 1.1% to 27% in Asia (China 73.8%, Hong Kong 69.6%). During the same period, in Europe, North America, and South America, the resistance to the same subtype increased only by 4.7%, 3.9%, and 4.3%, respectively. Despite that, over the years, the global resistance to amantadine rose to just over 12%. In most of the H3N2 isolates (approximately 98%), a single amino acid substitution (replacement of serine with asparagine localized at position 31 in M2; S31N) caused resistance to amantadine [15]. This mutation occurred in virus strains isolated across the world from humans, pigs, and birds. Sporadically, L26F, V27A, A30T, G34e, and L38F mutations were also detected. In comparison, between 2000 and 2004, only 0.3% of A/H1N1 strains showed resistance to amantadine. During the 2005/2006 season, the percentage of amantadine-resistant strains dramatically increased worldwide (90.6% for the A/H3N2 and 15.6% for the A/H1N1). Subsequent observations estimated that from 2013, approximately 45% of all circulating IAV subtypes were resistant to amantadine. The problem of mutations in the M2 protein mainly concerned strains with the following subtypes of haemagglutinin: H1 (69%), H3 (43%), H5 (28%), H7 (12%), and H9 (23%), but they did not occur in H8 and H12–16 variants.

The hemagglutinin mutations were also present in isolates (H17N10) obtained from bats. The vast majority of amantadine-resistant IAV subtypes (95%) had the S31N mutation. It was present in the following subtypes: over 96% of H1, 93% of H3, 83% of H5, 86% of H7, and 87% of H9. Other mutations occurred less frequently (L26F, V27A, A30T/V, G34E, and L38F). Double mutations were also observed in resistant strains, e.g., V27A + S31N and L26F + S31N. The adamantane-based compounds have been used not only in antiviral medication but also in drugs designed for the treatment of diabetes, Alzheimer’s, and Parkinson’s disease. Their ubiquitous use has caused the dramatic increase in resistance [34,35]. The phenomenon of increased resistance has also occurred in countries with low adamantanes consumption, showing a stable circulation of non-drug-pressure mutants with a replication and transmission capacity similar to a wild virus [34].

Therefore, according to the recommendations of the 2006 Advisory Committee on Immunization Practices in the USA and the Infectious Diseases Society of America (IDSA), the use of amantadine and rimantadine in influenza therapy is not currently recommended [36].

Mutations conferring amantadine and rimantadine resistance are described in Table 1 and Table 2.

### 2.2. Neuraminidase Inhibitors (NIs)

To infect the host’s cell, the virus uses its surface protein, hemagglutinin, to bind to the sialic acid located on the external surface of the host’s cell. After replication, the newly created viral particles remain attached to sialic acid groups [1]. Neuraminidase facilitates the release of viral particles by cleaving sialic acid groups. NIs are analogs of sialic acid. They act by binding to the cleavage site of the virus, which leads to the accumulation of the virions on the cell surface, and block its spread to the adjacent cells [42,43,44].

The idea of using a neuraminidase inhibitor as an antiviral agent sparked interest as early as 1948. It took almost twenty years before the first attempts were conducted (from 1966 to 1976). Unfortunately, the results showed low specificity and potency. The determination of the three-dimensional crystal structure and NA catalytic sites led to the design of the first two potent inhibitors. The first one, zanamivir (Relenza^®^), described in Table 3, was registered in 1999, followed by oseltamivir (Tamiflu^®^), described in Table 4, registered a year later. Two newer representatives of this group, and peramivir (Rapivab^®^, Rapiacta^®^, PeramiFlu^®^, Alpivab^®^), Table 5, laninamivir (Laninamivir^®^), Table 6, have been in use since 2010 and 2014, respectively.

Zanamivir and oseltamivir are prodrugs metabolized in vivo to their active form. They inhibit both IAV and IBV replication and reduce the duration time of the disease [33,34]. Studies demonstrated that neuraminidase inhibitors can decrease the duration of the illness by 30% and reduce the severity of the symptoms by approximately 40%, if they are administered within 36 hours from the symptoms’ onset. They also reduce the risk of incidence of the influenza complications, such as otitis media, sinusitis and pneumonia. Both zanamivir and oseltamivir can be used as a chemoprophylaxis. They decrease the risk of infection by approximately 70–90%, provided that they are administered shortly after exposure to IAV or IBV [15,56].

Zanamivir is a 4-deoxy-4-guanidino analog of DANA. It has a low oral bioavailability (2–10%), therefore it works best when administered by oral inhalation. This allows to deposit approximately 78% of the drug in the mouth and throat, which is the primary site of influenza virus replication [57]. According to European Medicines Agency zanamivir is indicated for the treatment of influenza A and influenza B in adults and children with influenza-like symptoms while the influenza virus is present in their community. Zanamivir is also indicated in adults and children (5 years of age) to prevent influenza A and influenza B that may develop in contact with clinically proven influenza in households. In exceptional circumstances, the use of zanamivir may be considered for the seasonal prevention of influenza A and B during an influenza epidemic (e.g., when there is a mismatch between circulating and vaccine strains, and in a pandemic situation). The use of antiviral drugs in the treatment and prevention of influenza should take into account official recommendations, the epidemiological factor, and the severity of the disease in particular geographical regions and patient populations. Zanamivir, given via oral inhalation, is recommended for treatment of influenza in patients aged 7 years or older, who have been symptomatic for no more than 2 days. Its 1% solution is licensed for usage as early as in children over 6 months old. Intravenous formulation of zanamivir (Dectova^®^, approved by EMA in 2019) can be used to treat critically ill patients with life-threatening infections caused by IAV or IBV, especially if they are resistant to other medications. Zanamivir is easily eliminated by kidneys in its unchanged form due to its hydrophilic properties [58,59,60]. Zanamivir effectively inhibits replication most of the oseltamivir-resistant viruses, including strains with one of the most common H275Y mutations in N1 viruses. A common mutation associated with zanamivir resistance in N2 viruses is E119G substitution; it confers high-level resistance to zanamivir. D and A substitutions were also reported; however, those mutations affect interaction with all NA inhibitors [61]. Other mutations which confer resistance to zanamivir are the I223R mutation in the A(H1N1)pdm09 strain and R292K NA in the A/H7N9 strain (both also result in reduced sensitivity to oseltamivir), and Gln136Lys (Q136K) NA mutation, resulting in a 70-fold reduction in zanamivir effectiveness. In IBV, the NA R371K mutation, NA G108E mutation, NA D197N, NA I221T, NA T146K, and NA T146P amino acid substitution led to a highly reduced sensitivity to both zanamivir and oseltamivir. A combination of T146P with a second mutation at N169S results in a further decline in the effectiveness of neuraminidase inhibitors. NA mutations H101L, A200T, D432G, H439P, and H439R also proved to decrease the effectiveness of neuraminidase inhibitors [40,47,48,49,50].

Another neuraminidase inhibitor is oseltamivir. This ethyl ester prodrug requires hydrolysis, mediated by esterases, to be converted to its active metabolite, oseltamivir carboxylate. According to EMA, oseltamivir is indicated in adults and children, including full-term neonates, who present with symptoms typical of influenza, when influenza virus is circulating in the community. Efficacy has been demonstrated when treatment is initiated within 2 days of first onset of symptoms, whereas recommended treatment time is 5 days. Oseltamivir shortens the duration of symptoms by 30% and reduces their severity from the first day of the therapy. EMA authorized oseltamivir as a post-exposure prevention in individuals 1 year of age or older following contact with a clinically diagnosed influenza case when influenza virus is circulating in the community. The use of oseltamivir for prevention should be determined on a case-by-case basis by the circumstances and the population requiring protection.

Oseltamivir’s bioavailability is significantly higher than zanamivir’s, reaching 80% [15,57]. The neuraminidase mutation related to the oseltamivir resistance was first detected in the A/H1N1 (A/Brisbane/59/2007-like) strain, which caused seasonal influenza in 2007. In oseltamivir-resistant A/H1N1 strains, histidine to tyrosine substitution at the aa 275th position (N1 numbering, H275Y) was detected. The H275Y mutation results in an approximately 400-fold reduction of the susceptibility of A/H1N1 to oseltamivir compared to the wild-type virus. At the turn of 2007/2008, H1N1 viruses with an H274Y mutation efficiently replicated and spread between people. Within weeks, the resistant viruses were detected in North America, Europe, and Asia. In certain countries, nearly 100% of A/H1N1 isolates were resistant to oseltamivir. Two years later, a new variant of the influenza virus, A/H1N1pdm09, emerged worldwide. In most cases, it remained oseltamivir-susceptible (<3% resistance). Since 2009, strains expressing resistance to oseltamivir were extremely rarely isolated [62,63,64,65]. The most commonly reported mutations that have appeared during treatment with oseltamivir in A/H3N2 viruses are E119V and R292K [66].

According to the Centers for Disease Control and Prevention (CDC; USA data) and Public Health England (PHE), 98.4%–100% of A/H1N1 strains isolated between 2013 and 2018 remained susceptible to oseltamivir. Moreover, all analyzed strains were also sensitive to zanamivir; however, they have retained a high-level resistance to adamantine [1].

Currently, the vast majority of A/H1N1 and A/H3N2 strains and IBV of the B/Victoria and B/Yamagata lines isolated from humans remain sensitive to all NIs. However, oseltamivir-resistant A/H1N1 strains can occur locally and are isolated from immunosuppressed individuals and critically ill patients. In those groups, detection of the H275Y strain containing A/H1N1 variants is associated with increased mortality. Studies demonstrated that treatment-related oseltamivir resistance is more frequent in the influenza A/H1N1 strain than in the A/H3N2 strain [67]. In A/H1N1 viruses, mutations involved in the reduction of susceptibility to zanamivir were detected in the Japanese pediatric population. The N70S NA mutation was responsible for a 46-fold increase in IC_50_ (half-maximal inhibitory concentration) for zanamivir, whereas the Q136K NA mutation resulted in a 300-fold increase. However, they were only detected in laboratory cultures, not in the clinical samples [63].

Both in vitro and in vivo studies have shown that influenza A/H3N2 strains and influenza B strains, which showcase a resistance to neuraminidase inhibitors, have lower replication capacity and transmission ability. In numerous cases, resistance can occur after prolonged treatment, usually longer than ten days, but in many cases, it can take more than a month for it to appear. These resistant variants often disappear once the treatment is stopped. All resistant A/H3N2 viruses had the R292K mutation related to the reduced susceptibility to oseltamivir and zanamivir. This dual resistance is extremely rare [15]. As of February 2017, a total of 14 influenza A and B phenotypes with reduced susceptibility to both oseltamivir and zanamivir had been identified [66].

In October 2009, the US Food and Drug Administration (FDA) authorized the use of peramivir (Rapivab^®^, Rapiacta^®^, PeramiFlu^®^, Alpivab^®^) for the emergency treatment of A/H1N1pdm09 pandemic influenza [68]. Five years later (2014), peramivir was approved for the treatment of acute, uncomplicated influenza in children (over 2 years of age) and in adults. In the same group of patients, peramivir is recommended by EMA. It should be given to the patient within two days from symptoms onset. Peramivir is administered as a single-dose intravenous infusion. It is not effective in preventing influenza, and, similar to other antiviral drugs, it cannot replace the annual vaccination. The studies have shown that peramivir resistance among influenza A/H1N1pdm09 viruses ranges from 1.3%–3.2% and is less than 1% for influenza A/H3N2 and B viruses. The most commonly occurring mutation leading to reduced effectiveness of peramivir is H274Y (H275Y N1 numbering), resulting in a 100–400-fold increase in IC_50_. It appears rapidly in immunocompromised individuals or hematopoietic cell transplant recipients treated with peramivir administered via the intravenous route. In vivo studies on mice showed that peramivir could be effective in the treatment of infections caused by oseltamivir-resistant A/H1N1/H274Y influenza virus. Human influenza B isolates with H273Y mutation in the NA active site have proven to be resistant to peramivir. Surveillance programs also detected several mutations, for example, I221V/T, D197N, and G104E (influenza B numbering), in the two IBV lineages (B/Victoria, B/Yamagata), which affected peramivir’s susceptibility. In vitro studies demonstrated that passaging of the A/H3N2 virus in the presence of peramivir led to the emergence of strains with reduced susceptibility to peramivir, which was related to the K189E mutation in the HA protein. The R294K mutation in the N9 neuraminidase resulting in high-level resistance to peramivir was detected in clinical strains [69,70].

Another neuraminidase inhibitor is laninamivir (Inavir^®^), or, more specifically, laninamivir octanoate. Currently, it is approved for use only in Japan. Laninamivir is a prodrug converted to its active metabolite by endogenous esterases in the airways.

In 2010, it was approved for the treatment of infections caused by IAV and IBV, and from 2013, it could also be used as post-exposure prophylaxis. Laninamivir is in the form of a powder and is administered via oral inhalation. The concentration of the drug in the respiratory tract exceeds the IC_50_ for influenza virus even up to 240 hours after inhalation. Its concentration in the respiratory tract was estimated to be 10,000 times higher than its concentration in plasma [57]. Laninamivir is an effective treatment for highly pathogenic avian influenza A/H5N1 and A/H1N1pdm09 viruses, as well as strains resistant to oseltamivir and zanamivir. In vitro studies showed that the drug remains in high concentration in the lungs for a long time. It binds to the viral neuraminidase more stably in comparison to other neuraminidase inhibitors. Its antiviral activity is achieved with a single dose (40 mg, via inhalation) [71,72]. The studies revealed numerous mutations leading to the emergence of influenza strains resistant to laninamivir.

The E119G mutation in N9 neuraminidase confers a 150-fold resistance to the drug. Another example is a double H275Y + I436N mutation, which reduces influenza virus susceptibility to laninamivir. Two variants of the A(H1N1)pdm09 strain significantly increased laninamivir IC_50_. The Q136K A and Q136R variants conferred a 25.5- and a 131.8-fold increase, respectively, in comparison to the wild-type. Furthermore, an NA N142S amino acid substitution in the A/H3N2 strain resulted in a 53-fold IC_50_ increase. Mutations decreasing laninamivir effectiveness were also detected in the IBV. They included T146P and the N169S mutation (in influenza B/Malaysia/RP2134/2019), resulting in an approximately 684-fold IC_50_ increase. The D197E mutation in the IBV conferred a 15-fold growth in laninamivir IC_50_ [45,48,50,73,74].

### 2.3. Viral Polymerase Inhibitors (PIs)

The influenza virus’s polymerase is composed of three subunits: PB1, PB2, and PA, which are imperative for viral RNA synthesis. A host pre-mRNA is used as a primer for the viral genome transcription. The transcription begins with a cap-snatching reaction, in which the host’s mRNA cap structure is bound by the cap-binding domain of PB2 and cleaved by the cap-dependent endonuclease of the PA subunit. This allows the virus to obtain short, capped primers derived from the host’s RNA, which then can be used for viral mRNA synthesis. The PB1 subunit is responsible for viral RNA elongation during transcription and replication. Polymerase inhibitors act by targeting viral polymerase subunits. Currently in use are two viral polymerase inhibitors: favipiravir and baloxavir.

Favipiravir inhibits PB1, whereas baloxavir blocks the PA subunit. Therefore, those therapeutic agents inhibit a process essential to the virus life cycle [75,76]. The PA subunit poses endonuclease activity. A characteristic feature of its inhibitors is their ability to chelate Mg^2+^ or Mn_2+_ ions in the enzyme’s catalytic site. This process is crucial for the enzyme’s activity. As previously mentioned, baloxavir inhibits the PA subunit; however, it is not the only compound capable of achieving that. Flavonoids were the subject of a recent study in which they were tested for their inhibitory potency. This study determined luteolin and its C-glucoside orientin to be potent inhibitors. A subsequent study focused on analyzing structural changes at C-7 and C-8 positions of luteolin. It determined that the presence of a 3′,4′-dihydroxyphenyl moiety is a critical feature for sub-micromolar inhibitory activity [77].

Baloxavir (Xofluza^®^) is a representative of a new class of anti-influenza drugs. This selective endonuclease inhibitor is a prodrug (baloxavir marboxil) hydrolyzed to its active metabolite, baloxavir acid, by arylacetamide deacetylase in the small intestine, blood, and liver. In 2018, baloxavir was approved for use in Japan, a year later in the US, and in the following years in Australia, Canada, and Switzerland. In July 2021, it was registered for therapy of acute uncomplicated influenza and post-exposure prophylaxis in the European Union [78]. Presently, the FDA has extended the indication for baloxavir use to patients with a high risk of developing influenza-related complications [79]. Baloxavir is approved for treatment of uncomplicated influenza in patients who are five years of age or older. The drug exhibited antiviral effects against both laboratory and clinical strains of IAV and IBV (including strains resistant to oseltamivir and types A/H5N1 and A/H7N9 [80,81]. Baloxavir is administered orally. It has a long half-life (80–100 h); thus, only a single dose is required for the drug to achieve its therapeutic efficacy. A single oral dose of baloxavir results in the drug reaching its peak plasma concentration after 4 hours in the fasted state [80,81,82,83].

Some influenza strains, mainly IAV, have expressed resistance to baloxavir. It is a result of the mutation associated with the replacement of isoleucine at a position of 38 by tyrosine, methionine, or phenylalanine in the PA subunit of RNA polymerase. This substitution leads to changes in the spatial conformation of the PA subunit, which makes it more difficult for baloxavir to attach itself to the virus [83].

Hayden et al. reported the emergence of baloxavir-resistant viruses in up to 9.7% of adult clinical trial participants and 23.4% of children baloxavir recipients [84]. Its reduced susceptibility results from PA I38T/M/F substitutions. There are 19 possible amino acid (AA) substitutions at polymerase acidic (PA) 38. Only three of them (R, K, and P) decreased polymerase activity to less than 79%. In comparison to the prototypical baloxavir marboxil resistance marker T38, five substitutions (M, L, F, Y, and C) resulted in 10%–35% reductions in baloxavir acid inhibitory activity, and 11 substitutions conferred more than 50% reductions in the drug’s effectiveness (R, K, S, N, G, W, A, Q, E, D, and H). The two remaining substitutions (V and P) did not decrease baloxavir inhibitory activity. From 2016 to 2018, before the baloxavir approval, a small percentage of viruses with mutations at PA residue 38 were detected. A resistant A/H3N2 virus was isolated from a pediatric patient who had not been previously exposed to baloxavir. Therefore, there is a danger that a pre-existent resistance to baloxavir can spread among humans, along with the increasing usage of this drug in a clinical setting [85,86]. Mutations conferring baloxavir resistance are described in Table 7.

Favipiravir (Avigan^®^) was registered in 2014 in Japan for the treatment of influenza caused by new and remerging pandemic viruses. Similarly to baloxavir, favipiravir is also a prodrug. It is a purine nucleotide that undergoes intracellular phosphorylation to become its active form, favipiravir-ribofuranosyl-5’-triphosphate (favipiravir-RTP). It is a competitive inhibitor of RNA-dependent RNA polymerase. Favipiravir’s bioavailability after oral administration reaches over 97%. The drug is well tolerated and shows activity against influenza type A, B, and C viruses. Not only does it inhibit the proliferation of strains that are causing seasonal infections (A/H1N1, A/H1N1pdm09, A/H3N2, and influenza B strains), but it also inhibits a highly pathogenic avian flu A/H5N1 strain isolated from humans [90,91].

Thus far, reduced susceptibility to favipiravir in clinical strains of the influenza virus has not been reported. The resistance to this drug in the pandemic A/H1N1 has been reported only in laboratory influenza strains. To elicit a robust resistance to favipiravir, a combination of two mutations is necessary. The first one is the K229R mutation in the PB1 subunit, which stops the integration of favipiravir into viral RNA by polymerases from H1N1, H3N2, and H7N9 influenza A viruses. The second one is a P653L mutation in the PA subunit, which counteracts the decreased RNA polymerase activity in cell culture resulting from the K229R mutation. The K229R and P653L mutations led to the emergence of a virus which was 30-fold less susceptible to favipiravir in comparison to the wild-type virus [92,93]. Mutations conferring favipiravir resistance are described in Table 8.

## 3. Future of the Influenza Treatment

The emergence of antiviral-drug-resistant influenza strains resulted in an intense search for an alternative treatment. A potential solution could be monoclonal antibodies. Its mechanism of action differs from traditional polyclonal antibodies induced by vaccination [94,95].

The VIS410 is an example of an IgG1 monoclonal antibody that has displayed, both in vitro and in vivo, activity against the influenza virus. A phase 2a clinical trial was conducted to assess its safety and tolerability in patients with uncomplicated influenza. It determined that VIS410 was safe and well tolerated in adults with uncomplicated influenza A and had a favorable effect on symptoms resolution and virus replication [96,97]. Another trial was conducted to assess the efficiency and safety of VIS410 in combination with oseltamivir vs. oseltamivir alone. The results of this trial are still under quality control review [98]. Another antibody considered for anti-influenza therapy is the MHAA4549A. A phase 2 randomized trial was conducted to determine its safety, tolerability, efficacy, and pharmacokinetics in subjects with uncomplicated influenza A infection. It compared adult patients testing positive for influenza A who were given a single intravenous dose of MHAA4549A with individuals who received an intravenous placebo. There were no statistically significant differences between the two groups in the average duration of the symptoms, nasopharyngeal viral load, or duration of viral shedding. There is one more antibody that has the potential to join the anti-influenza armory. The CT-P27 is a combination of two other antibodies: CT120 and CT149. It displayed therapeutic efficacy, long prophylactic potency, and synergistic effect with oseltamivir in in vivo studies on mouse models [94,95,97,99].

Another potential candidate for a new anti-influenza drug is Ro-3306, a selective ATP-competitive inhibitor of cyclin dependent kinase 1 (CDK1). This CDK1 inhibitor displayed substantial antiviral activity against all tested influenza viruses without having a cytotoxic effect on the host’s cell. It was also proven to be effective against influenza strains resistant to oseltamivir [100].

The lack of optimal influenza treatment turned scientists’ attention to natural therapies. Several herbal remedies were tested to assess their potential to prevent and treat viral respiratory illnesses. For years, scientists have known that the intestinal microbiome plays a crucial role in pulmonary immunity and defense against viral respiratory infections. Influenza can affect the composition of the gut’s microbiota, thus aggravating the severity of the illness [101]. Therefore, probiotics, which are live nonpathogenic microorganisms given to individuals to improve the gut microbiome, are considered a potential anti-influenza treatment. Their effectiveness appears to be significantly dependent on the type of virus, the type of administered probiotics, dosage, and treatment duration. Studies have suggested that probiotics may improve vaccine efficiency, as well as help to reduce viral pathogenicity. However, it is necessary to emphasize that no direct effects of probiotics on viral pathogenic mechanisms have been proven so far, and their mechanism of action is yet to be explored [64,102,103].

The search for antiviral drugs has also shown that numerous compounds of natural origin, as well as their synthetic derivatives, might potentially be useful in the development of new antiviral drugs [104,105,106]. An example of such a compound is Maoto, a multicomponent formulation extracted from crude ephedra herb, apricot kernel, cinnamon bark, and glycyrrhiza root. Maoto is used as a Japanese traditional herbal medicine. Studies conducted on a mouse model suggested that it has an anti-inflammatory effect mediated via β-adrenergic receptors, although its exact mechanism of action remains unclear [104]. Another example of an herbal remedy is liquorice roots, containing glycyrrhizin, which stimulates T cells to produce interferon-gamma. Berries’ extracts and pomegranate, both containing polyphenols, are considered to improve T cells function and reduce viral replication. Similarly, *Epimedium koreanum* is also believed to suppress influenza virus replication; it might enhance the secretion of type I interferon and proinflammatory cytokines. Also under consideration is *Clinacanthus siamensis,* which enhances the production of anti-influenza virus antibodies. *Psidium guajava* seems to block viral hemagglutinin, and *Scutellaria baicalensis* is considered to inhibit virus budding [105].

## 4. Conclusions

Genetic variation is a typical feature of all living organisms and viruses. It is the foundation for natural selection and adaptation to a changing environment. In RNA viruses, there is a higher mutation rate compared to DNA viruses. This is because of the absence of specific mechanisms in RNA polymerase to check and correct the nucleotide attachment during replication.

The accumulation of molecular changes in eight influenza RNA segments occurs through point mutations, genetic reassortment, defective interfering molecules, and RNA recombination, and can lead to large changes in the biological properties of influenza viruses, including changes in the virulence, adaptation to new hosts, and resistance to antiviral drugs, avoiding the immune response of an infected organism. Virus variability hinders the prevention of influenza by implementing an effective preventive program in the form of vaccinations. In theory, the gene reassortment might lead to production of as many as 256 genetically different types of viral progeny. A new variant may cause a pandemic. Therefore, influenza viruses are considered a real threat to public health. The last pandemic occurred in 2009; the A/(H1N1)pdm09 was a quadruple reassortant with a genome composed of two swine-origin viruses, one avian-origin virus, and one human-origin virus. This genetic incident highlighted the threat caused by the influenza virus and limited possibilities to control infections.

Since then, several antiviral drugs belonging to the class of neuraminidase inhibitors and viral polymerase inhibitors have been synthesized, tested, and approved. They have replaced M2 protein inhibitors, which have not been recommended to treat influenza since 2006 because of the occurrence of mutations conferring amantadine and rimantadine resistance in strains circulating worldwide. At present, the priority is to conduct rational antiviral therapy to reduce the risk of selecting resistant strains and to make efforts to monitor influenza virus resistance by the national reference laboratories.

## Figures and Tables

**Table 1 ijms-23-12244-t001:** Chemical name and structure of amantadine and mutations conferring resistance to M2 protein inhibitor.

AMANTADINE				
**IUPAC Name** adamantan-1-amine		**Chemical Formula** 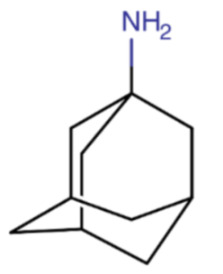		
**Strains**	**Year/Period** **of Time**	**Mutations Conferring** **Resistance **	**Region** **of the World**	**Ref.**
A(H1N1) A(H1N2) A(H3N2)	1981–2001	G16E, S31N, R77Q, R18K, C19Y, D21G, S22L, P25L, V27A, F485, R54H, R61K	Germany	[37]
A(H3N2)	1995–2005	L26F, V27A, A30T, S31N	Asia, USA	[38]
A(H3N2)	2004–2005	S31N mutation	Asia, Europe, Australia, USA	[39]
A(H1N1)pdm09	2009	S31N polymorphism		
A(H7N9)	2013	S31N	China	[40]
A(H1N1) A(H1N2) A(H3N2)	N/D	V27A, S31N	N/D	[37]
A(H1N1)	N/D	L26F, V27A, A30T, S31N, G34E, V27A/S31N	N/D	
A(H1N1)pdm09	2011/2012 2014/2015	V27A/S31N	Iran	[41]
A(H3N2)	2011/2012 2014/2015	V27A/S31N	Iran	[41]

N/D: no data.

**Table 2 ijms-23-12244-t002:** Chemical name and structure of rimantadine and mutations conferring resistance to M2 protein inhibitor.

RIMANTADINE				
**IUPAC Name** 1-(1-adamantyl)ethanamine		**Chemical Formula** 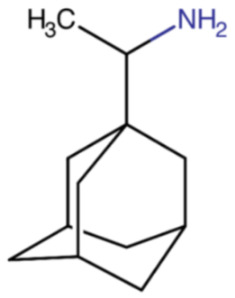		
**Strains**	**Year/Period** **of Time**	**Mutations** **Conferring** **Resistance**	**Region** **of the World**	**Ref.**
A(H3N2)	1995–2005	L26F, V27A, A30T, S31N	Asia, USA	[38]
A(H7N9)	2013	S31N	China	[40]

**Table 3 ijms-23-12244-t003:** Chemical name and structure of zanamivir and mutations conferring resistance to neuraminidase inhibitor.

ZANAMIVIR				
**IUPAC Name** (2R,3R,4S)-3-acetamido-4-(diaminomethylideneamino)-2-[(1R,2R)-1,2,3-trihydroxypropyl]-3,4-dihydro-2H-pyran-6-carboxylic acid		**Chemical Formula** 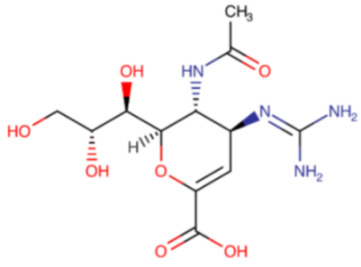		
**Strains**	**Year/Period** **of Time**	**Mutations Conferring** **Resistance**	**Region** **of the World**	**Ref.**
A(H1N1)	N/D	E119V, H126N, Q136K	N/D	[39]
H3N2	N/D	E119V	N/D	[4]
A(H1N1)pdm09	2012–2022	Q136L/R/K	Asia–Pacific region	[45]
A(H3N2)	2013–2022	Q136L/K	Asia–Pacific region	
A(H1N1)	2006–2008	Q136K	Australasia, Southeast Asia	[46]
A(H7N9)	2013	R292K	China	[40]
A(H1N1)pdm09	2014	I223R	Denmark	[47]
A(H1N1)pdm09	2014–2015	I223R	Bolivia	[48]
A(H3N2)	2014–2015	Q136K(NA)N142S(NA)	Western Pacific, Europe, USA	[48]
Influenza B viruses	2004–2005	R371K	Hong Kong	[49]
Influenza B viruses	2014–2015	D197N(NA)I221T(NA)	Australia, USA, China, Ukraine, Japan, Russia	[48]
Influenza B viruses	2018–2020	T146K	Philippines	[50]
		T146P, N169S, G247D, I361V, G108E, H101L, A200T, H439P	Malaysia	

N/D: no data.

**Table 4 ijms-23-12244-t004:** Chemical name and structure of oseltamivir and mutations conferring resistance to neuraminidase inhibitor.

OSELTAMIVIR				
**IUPAC Name** ethyl (3R,4R,5S)-4-acetamido-5-amino-3-pentan-3-yloxycyclohexene-1-carboxylate		**Chemical Formula** 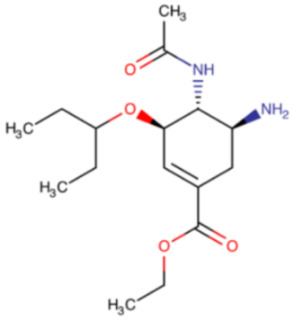		
**Strains**	**Year/Period** **of Time**	**Mutations Conferring** **Resistance**	**Region** **of the World**	**Ref.**
A(H1N1)	2005–2008	H274Y	Germany	[51]
A(H3N2)		E119V, R292K, N294S		
A(H5N1)		H274Y		
A(H1N1)	2009	I223V	USA	[52]
A(H1N1)	2007–2008	H274Y	Oceania, South Africa, South East Asia	[53]
A(H1N1)	2007–2008	H275Y	Europe, Asia, USA	[39]
A(H1N1)pdm09	2009–2016	H275Y	Germany	
A(H3N2)	2012	R292K	Germany	
A(H1N1)pdm09	2013–2014	H275Y	China, Japan, USA	
A(H1N1)pdm09	2014	H275Y, H275Y/G147R, I223R	Denmark	[47]
A(H1N1)pdm09	2014–2015	H275Y(NA)I223R(NA)	Australia, Hawaii, Ukraine, France, Bolivia	[46]
A(H3N2)	2014–2015	I222T/S331R E119V(NA) R292K(NA), G320E(NA) N142S(NA)	Western Pacific, Americas, Europe USA	[48]
A(H1N1)pdm09	2009–2010	H275Y(NA)	Greece	[54]
A(H7N9)	2013	R292K	China	[40]
Influenza B viruses	2004–2005	R371K	Hong Kong	[49]
Influenza B viruses	2011	H273Y	Canada	[55]
Influenza B viruses	2014–2015	I221T(NA) K152M(NA) D197N(NA) I221T(NA)	Honduras Bangladesh Australia, USA, China, Ukraine Japan, Russia, China	[48]
A(H3N2)	2011–2013	Q136L, Q136K	Asia–Pacific region	[45]
Influenza B viruses	2016–2019	T146K, T146P, T146P, N169S, G108E, A200T	Asia	[50]

N/D: no data.

**Table 5 ijms-23-12244-t005:** Chemical name and structure of peramivir and mutations conferring resistance to neuraminidase inhibitor.

PERAMIVIR				
**IUPAC Name** (1S,2S,3S,4R)-3-[(1S)-1-acetamido-2-ethylbutyl]-4-(diaminomethylideneamino)-2-hydroxycyclopentane-1-carboxylic acid		**Chemical Formula** 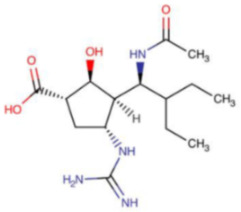		
**Strains**	**Year/Period of Time**	**Mutations Conferring** **Resistance**	**Region** **of the World**	**Ref.**
A(H1N1)	2006–2008	Q136K(NA)	Australasia, Southeast Asia	[46]
A(H1N1)pdm09	2011–2012	Q136R(NA), Q136K(NA)	Asia–Pacific region	[45]
A(H1N1)pdm09	2014	H275Y/G147R	Denmark	[47]
A(H1N1)pdm09	2014–2015	H275Y(NA)	Australia, Hawaii, Ukraine, France	[48]
A(H3N2)	2014–2015	E119V(NA), R292K(NA) N142S(NA)	Japan USA	[48]
Influenza B viruses	2014–2015	T106P, G104R/G, G145E I221T(NA) D197N(NA) I221T(NA)	Japan Honduras Australia, USA, China, Ukraine Japan, Russia, China	[48]
A(H1N1)pdm09	2009–2012	H275Y(NA)	Asia, Africa, Oceania	[40]
Influenza B viruses		I221T, A245T, K360E, A395E, D432G, G145R + Y142H		
Influenza B viruses	2011	H273Y	Canada	[40]
A(H7N9)	2013	R292K	China	[55]
Influenza B viruses	2016–2020	T146K, T146P, N169S, G247D, I361V, G108E, H101L, A200T, D432G, H439P	Asia	[50]

**Table 6 ijms-23-12244-t006:** Chemical name and structure of laninamivir and mutations conferring resistance to neuraminidase inhibitor.

LANINAMIVIR				
**IUPAC Name** (2R,3R,4S)-3-acetamido-4-(diaminomethylideneamino)-2-[(1R,2R)-1,2,3-trihydroxypropyl]-3,4-dihydro-2H-pyran-6-carboxylic acid		**Chemical Formula** 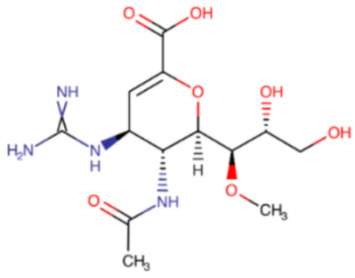		
**Strains**	**Year/Period** **of Time**	**Mutations Conferring** **Resistance **	**Region** **of the World**	**Ref.**
A(H1N1)pdm09	2011–2012	Q136R/K	Asia–Pacific region	[45]
H3N2 wt	E119G	N/D	N/D	[50]
A(H3N2)	2014–2015	N142S(NA)	USA	[48]
Influenza B viruses	2019	T146P(NA), N169S(NA) T146K	Malaysia	[50]
	2018	T146K	Philippines	
	2019	A200T	Malaysia	

N/D: no data.

**Table 7 ijms-23-12244-t007:** Chemical name and structure of baloxavir marboxil and mutations conferring resistance to polymerase inhibitor.

BALOXAVIR MARBOXIL				
**IUPAC Name** [(3R)-2-[(11S)-7,8-difluoro-6,11-dihydrobenzo[c][1]benzothiepin-11-yl]-9,12-dioxo-5-oxa-1,2,8-triazatricyclo [8.4.0.03,8]tetradeca-10,13-dien-11-yl]oxymethyl methyl carbonate		**Chemical Formula** 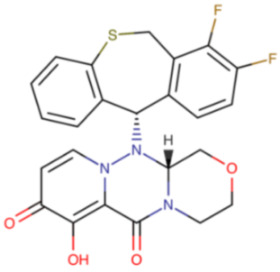		
**Strains**	**Year/Period** **of Time**	**Mutations Conferring** **Resistance**	**Region** **of the World**	**Ref.**
A(H1N1)pdm09	2018	PA-I38T, PA-I38F, PA-I38M	USA	[84]
A(H3N2)	2018/2019 2018	PA-I38T PA-I38M, PA-I38F	Japan	[87,88]
Influenza B viruses	2020	PA-I38T, PA-I38M	Reported only in vitro	[89]

**Table 8 ijms-23-12244-t008:** Chemical name and structure of favipiravir and mutations conferring resistance to polymerase inhibitor.

FAVIPIRAVIR				
**IUPAC Name** 5-fluoro-2-oxo-1H-pyrazine-3-carboxamide		**Chemical Formula** 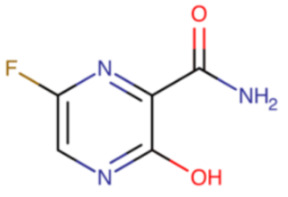		
**Strains**	**Year/Period** **of Time**	**Mutations Conferring** **Resistance**	**Region** **of the World**	**Ref.**
A(H1N1)pdm09 A(H3N2), A(H7N9)	2018	K229R P653L	Reported only in vitro	[93]

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
