# Peer review of "Evolution of Influenza Viruses—Drug Resistance, Treatment Options, and Prospects"

_ijms, 2022, doi:10.3390/ijms232012244_

Round 1

Reviewer 1 Report

The presented review is focusing on the history of influenza viruses, pandemics and anti-viral treatment. Especially, the influenza inhibitors are nicely described. The review is well written. I have some minor comments:

-          NS2 protein is nuclear export protein (NEP)

-          P.2. line 85 – M2 protein should be mention first time in full name as envelope matrix protein 2 (M2), line 93, should be mention only as M2 protein and the function –  „proton- selective viroporine and  plays a part in the uncoating and assembling of the virus particles“  should be included into the text.

-          The sentence (line 95-96) can be deleted.

-          It will be appropriate to include also information, recommendation, approval and decision about influenza inhibitors from European medicines agency (EMA).

-          Line 499 the ant-influenza armory – should be corrected (spelling)

-          What kind of inhibitor is RO-3306. Please include basic information.

Author Response

We would like to thank the Reviewer for their thorough review of the manuscript and for comments and suggestions that helped us improve the manuscript. We highlight all manuscript corrections.

NS2 protein is nuclear export protein (NEP)
This information has been added. Please see line 94.

P.2. line 85 – M2 protein should be mention first time in full name as envelope matrix protein 2 (M2), line 93, should be mention only as M2 protein and the function –  „proton- selective viroporine and plays a part in the uncoating and assembling of the virus particles“  should be included into the text.
These changes have been applied. Please see lines 89-91 and 98 and 99.

The sentence (line 95-96) can be deleted.
We deleted the sentence.

It will be appropriate to include also information, recommendation, approval and decision about influenza inhibitors from European medicines agency (EMA).
We introduced the recommendations of the EMA. Please see the text, lines 258-265, 271-273, 291-300, 350.

Line 499 the ant-influenza armory – should be corrected (spelling)
Corrected.

What kind of inhibitor is RO-3306. Please include basic information.
Information included (lines 499-500)

We wrote the IC50 with subscript, but the font used justifies the look of the notation. The table and the titles have been matched. 

Reviewer 2 Report

The manuscript "Evolution of influenza viruses - drug resistance, treatment options and prospects" addresses an important issue: the increasing resistance to antiviral treatments by influenza viruses worldwide. In this article, the authors discuss problems resulting from the rapid evolution of the influenza virus and its growing resistance to the available antiviral drugs.

The objectives were clearly stated and explained in the manuscript, and the experimental strategy was appropriate to gather the experimental information from which the conclusions were drawn. The manuscript is well written and has good organization. However, the examination of the experimental data and the discussion of the results is lacking a more in-depth analysis. The authors should consider different alternative explanations/considerations for interpreting the results and should provide conclusions to their investigation.

Some major points deserve careful attention:

1.      The authors should discuss more extensively on the conclusions extracted from their work in the Conclusions section of the manuscript. Only one paragraph is not enough and it really doesn’t provide any relevant information as is.

2.      In the Introduction section the authors may address the effect of EAAs in other physiological processes better understood such as muscular growth more in depth as this may help clarifying some of the effects seen in other physiological processes such as those related to bone homeostasis.

3.      In the Abstract section the authors should address more thoroughly the aim of the study and revise the English writing of the entire manuscript as there are many spelling and grammar errors.

Minor points:

1.      The “50” in IC50 should be in subscript not just in a lower font size.

2.   Caption in all the tables should be extended to become more comprehensive.

3.      Figure’s size inside all the tables should be rechecked and adapted when needed.

Author Response

We thank for careful review our manuscript, which helped to improve the quality of the paper. We highlight all  corrections.

The authors should discuss more extensively on the conclusions extracted from their work in the Conclusions section of the manuscript. Only one paragraph is not enough and it really doesn’t provide any relevant information as is.
The Conclusions section has been completed.

 In the Introduction section the authors may address the effect of EAAs in other physiological processes better understood such as muscular growth more in depth as this may help clarifying some of the effects seen in other physiological processes such as those related to bone homeostasis.
The aim of the work is to describe the evolutionary changes of the virus, which also includes resistance to direct antiviral drugs. It seems to us that information on EAAs supplementation, go beyond the scope of the study.

In the Abstract section the authors should address more thoroughly the aim of the study and revise the English writing of the entire manuscript as there are many spelling and grammar errors.
The Abstract section has been edited. All work has been checked, linguistic and punctuation errors are corrected.

The “50” in IC50 should be in subscript not just in a lower font size.
We wrote the IC50 with a subscript, but the font used justifies the look of the notation.

Caption in all the tables should be extended to become more comprehensive. Figure’s size inside all the tables should be rechecked and adapted when needed.
The tables and captions have been matched.